

# Significant concentrations of nitryl chloride sustained in the morning: Investigations of the causes and impacts on ozone production in a polluted region of northern China

Yee Jun Tham[1], Zhe Wang[1], Qinyi Li[1], Hui Yun[1], Weihao Wang[1], Xinfeng Wang[2], Likun Xue[2], Keding Lu[3], Nan Ma[4], Birger Bohn[5], Xin Li[5], Simonas Kecorius[4] and Johannes Größ[4], Min Shao[3], Alfred Wiedensohler[4], Yuanhang Zhang[3], and Tao Wang[1*].

[1]Department of Civil and Environmental Engineering, The Hong Kong Polytechnic University, Hong Kong, China,
[2]Environment Research Institute, Shandong University, Jinan, Shandong, China,
[3]State Key Joint Laboratory of Environmental Simulation and Pollution Control, College of Environmental Sciences and Engineering, Peking University, Beijing, China,
[4]Leibniz Institute for Tropospheric Research, Permoserstr. 15, 04318 Leipzig, Germany,
[5]Forschungszentrum Jülich, Institut IEK-8: Troposphäre, 52425 Jülich, Germany.

*Correspondence to: T. Wang (cetwang@polyu.edu.hk)

**Abstract.** Nitryl chloride ($ClNO_2$) is a dominant source of chlorine radical in polluted environment and can significantly affect the atmospheric oxidative chemistry. However, the abundance of $ClNO_2$ and its exact role are not fully understood under different environmental conditions. During the summer of 2014, we deployed a chemical ionization mass spectrometer to measure $ClNO_2$ and dinitrogen pentoxide ($N_2O_5$) at a rural site in the polluted North China Plain. Elevated mixing ratios of $ClNO_2$ (>350 pptv) were observed at most of the nights with low levels of $N_2O_5$ (<200 pptv). The highest $ClNO_2$ mixing ratio of 2070 pptv (1-min average) was observed in a plume from megacity (Tianjin) and was characterized with faster $N_2O_5$ heterogeneous loss rate and $ClNO_2$ production rate compared to average condition. The abundant $ClNO_2$ concentration kept increasing even after sunrise and reached a peak 4 hours later. Such highly sustained $ClNO_2$ peaks after sunrise are discrepant from the previously observed typical diurnal pattern. Meteorological and chemical analysis show that the sustained $ClNO_2$ morning peaks are caused by significant $ClNO_2$ production in the residual layer at night followed by downward mixing after break-up of the nocturnal inversion layer in the morning. We estimated that ~1.7-4.0 ppbv of $ClNO_2$ would exist in the residual layer in order to maintain the observed morning $ClNO_2$ peaks at the surface site. Observation-based box model analysis show that photolysis of $ClNO_2$ produced chlorine radical with a rate up to 1.12 ppbv h$^{-1}$, accounting for 10-30% of primary $RO_x$ production in the morning hours. The perturbation in total radical production leads to an increase of integrated daytime net ozone production by 3% (4.3 ppbv) on average, and with a larger increase of 13% (11 ppbv) in megacity outflow that was characterized with higher $ClNO_2$ and relatively lower OVOC to NMHC ratio.



# 1 Introduction

Nitryl chloride ($ClNO_2$) is a nocturnal reservoir of reactive nitrogen and chlorine radicals (Cl) that play crucial roles in the next day photochemistry (Young et al., 2012; Mielke et al., 2013; Sarwar et al., 2014). Formation of $ClNO_2$ begins with the oxidation of nitrogen dioxide ($NO_2$) by ozone ($O_3$) to yield nitrate radical ($NO_3$) (R1). $NO_3$ is very susceptible to sunlight and can react rapidly with nitrogen oxide (NO) and volatile organic compounds (VOCs) (R2-4). At nightfall, the $NO_3$ begin to accumulate and can further react with $NO_2$ to give $N_2O_5$ (R5).

$$NO_2 + O_3 \rightarrow NO_3 \tag{R1}$$
$$NO_3 + h\nu \rightarrow 0.9NO_2 + 0.9O + 0.1NO + 0.1O_2 \tag{R2}$$
$$NO_3 + NO \rightarrow 2NO_2 \tag{R3}$$
$$NO_3 + VOC \rightarrow products \tag{R4}$$
$$NO_3 + NO_2 + M \leftrightarrow N_2O_5 + M \tag{R5}$$

$N_2O_5$ exist in thermal equilibrium with $NO_2$ and $NO_3$, and heterogeneously reacts with chloride containing aerosols ($Cl^-$) to form $ClNO_2$ and nitrates ($NO_3^-$) (R6), or undergoes hydrolysis to produce water-soluble nitric acids ($HNO_3$) (R7) (Finlayson-Pitts et al., 1989).

$$N_2O_5 + Cl^- (aq) \rightarrow ClNO_2 + NO_3^- (aq) \tag{R6}$$
$$N_2O_5 + H_2O (l) \rightarrow 2HNO_3 \tag{R7}$$

The abundance of $ClNO_2$ produced from the heterogeneous uptake of $N_2O_5$ depends on the availability of $Cl^-$ aerosols and nitrogen oxides ($NO_x = NO + NO_2$) in the atmosphere (Bertram and Thornton, 2009; Brown and Stutz, 2012). Efficient production of $ClNO_2$ was found in the polluted coastal regions that are directly impacted by abundant of sea salt aerosol and urban emissions. For instance, Osthoff et al. (2008) measured more than 1 ppbv of $ClNO_2$ in the urban outflows along the coast of Texas; high $ClNO_2$ mixing ratios of up to 3.6 ppbv were detected in the polluted Los Angeles basin (Riedel et al., 2012; Wagner et al., 2012; Mielke et al., 2013). Significant production of $ClNO_2$ was not previously expected in inland regions with limited $Cl^-$ sources until Thornton et al. (2010) found $ClNO_2$ mixing ratio of up to 0.45 ppbv in urban plumes of Boulder, Colorado. They implied the presence of abundant non-oceanic chloride from coal-fired power plants, industries, biomass burning, road salts and soil-dust in inland regions which could support widespread production of $ClNO_2$. Since then, other studies have observed $ClNO_2$ mixing ratios, ranging from tenths of pptv up to 1.3 ppbv in polluted inland regions (Mielke et al., 2011; Phillips et al., 2012; Riedel et al., 2013; Faxon et al., 2015).

$ClNO_2$ is unreactive during the night as it has negligible nocturnal loss processes. Its primary sink is via photolysis during the day, yielding a highly reactive chlorine radical and $NO_2$ (R8).





$$ClNO_2 + hv \rightarrow Cl + NO_2 \quad\quad\quad (R8)$$

Therefore, $ClNO_2$ typically show a clear diurnal pattern where it accumulates primarily at night and decreases gradually to very low concentrations in the daytime. Under conditions of reduced photolysis, small $ClNO_2$ concentrations may persist during the daytime. For example, Mielke et al., (2013) found that the median mixing ratios of $ClNO_2$ could reach 100 pptv

even 4 hours after sunrise at Pasadena, which was partly caused by the heavy cloud and aerosol cover or fog at the site. More recent measurements on surface sites at London and Texas observed cases with increase of $ClNO_2$ soon after sunrise and peak several hours later with concentration of 40-150 pptv that could result from transport processes of $ClNO_2$ from region with higher $ClNO_2$ concentrations (Bannan et al., 2015; Faxon et al., 2015).

Photolysis of nighttime accumulated $ClNO_2$ during daytime was found to cause rapid production of Cl, with peak of 0.03 - 0.50 ppbv $h^{-1}$ (Osthoff et al., 2008; Thornton et al., 2010; Phillips et al., 2012; Riedel et al., 2012, Riedel et al., 2014; Mielke et al., 2015). This Cl precursor was shown to be an important primary radical source as it constituted ~9-13% of the daily primary radical productions (Edwards et al., 2013; Young et al., 2014) and exceeded the production of hydroxyl radical (OH) via photolysis of $O_3$ by up to a factor of 10 for several hours after sunrise (Phillips et al., 2012). Subsequently, the

released Cl would oxidize VOCs and enhance ozone production in polluted regions through reactions of R9-R15.

$$Cl + RH \rightarrow R + HCl \quad\quad\quad (R9)$$
$$R + O_2 + M \rightarrow RO_2 + M \quad\quad\quad (R10)$$
$$RO_2 + NO \rightarrow RO + NO_2 \quad\quad\quad (R11)$$
$$RO + O_2 \rightarrow OVOC + HO_2 \quad\quad\quad (R12)$$

$$HO_2 + NO \rightarrow OH + NO_2 \quad\quad\quad (R13)$$
$$NO_2 + hv \rightarrow NO + O(^3P) \quad\quad\quad (R14)$$
$$O(^3P) + O_2 + M \rightarrow O_3 + M \quad\quad\quad (R15)$$

For example, Osthoff et al. (2008) reported an increase of 6 and 9 ppbv of ozone in Houston by constraining 0.65 and 1.5 ppbv of $ClNO_2$ into their model, respectively. Neglecting the contribution of HONO, 1.5 ppbv of $ClNO_2$ could increase ~12 ppbv

of ozone in Los Angeles (Riedel et al., 2014). Chemical transport model simulations by Sarwar et al. (2014) suggested that high $ClNO_2$ concentrations in region like China and Western Europe can lead to daily 8-hour average ozone increase of up to 7 ppbv.

Despite the important role in photochemistry, studies on this Cl precursor in China are sparse. Most of the previously

reported studies of $ClNO_2$ were conducted in the US (e.g. Osthoff et al., 2008; Thornton et al., 2010; Riedel et al., 2012; Riedel et al., 2013; Mielke et al., 2013; Faxon et al., 2015), Canada (Mielke et al. 2011, 2015), with a few in Europe (Phillips et al., 2012; Bannan et al., 2015). Recent measurements in Hong Kong of southern China have observed high levels of $ClNO_2$ at both surface and mountain sites (Tham et al., 2014; X. Wang et al., 2014; Wang et al., 2016). In a well-processed regional plume, maximum $ClNO_2$ of 4.7 ppbv and $N_2O_5$ of up to 7.7 ppbv were observed at Mt Tai Mao Shan (957 m a.s.l.), and box



model calculations showed significant impacts of the $ClNO_2$ on the next day ozone production with an increase of up to 41% (Wang et al., 2016).

The North China Plain (NCP) covers an area of 409,500 $km^2$ and is home to megacities like Beijing, Tianjin, and Shijiazhuang. It is one of the most polluted regions in China according to the Ministry of Environmental Protection (MEP China, 2015). Due to intense and fast economic development, the emission of $NO_x$ has increased steadily, reaching a peak of 127 Gg N $yr^{-1}$ in 2011 (Mijling et al., 2013). High levels of ground-level $O_3$ were frequently reported in the NCP. For instance, a maximum hourly value of up to 286 ppbv was observed in a rural site north of Beijing (Wang et al., 2006). Tropospheric ozone over the last two decades has increased at a rate of 2-5% $yr^{-1}$ (Ding et al., 2008; Zhang et al., 2014). The abundant $NO_x$ and $O_3$ coupled with the large loading of chloride-containing aerosol (Sun et al., 2006; Huang et al., 2014; Sun et al., 2015) may make the heterogeneous uptake and chlorine activation processes particularly important in driving the formation of ozone and secondary aerosol in this region.

In summer 2014, we deployed a chemical ionization mass spectrometer (CIMS) for the first field measurement of $ClNO_2$ in the NCP. It was a part of an international collaborative field campaign, the CARE-Beijing 2014 (Campaigns of Air Quality Research in Beijing and Surrounding Regions) with the major aim to understand the oxidative processes in the region. In the present paper, we give an overview of the measurement results of $ClNO_2$ and its precursors, $N_2O_5$, and related species. We then examine the factors that affect the $ClNO_2$ production. We also investigate the cause of sustained $ClNO_2$ peaks observed after sunrise and the potential sources of aerosol chloride that drive the $ClNO_2$ productions. The impacts of the $ClNO_2$ on the primary radical productions and ozone formation are then assessed.

## 2 Methodology

### 2.1 Site description

This study took place at a semi-rural site (38.665° N, 115.204° E) in Wangdu County of Hebei province. Figure 1 shows the location of the measurement site in relation to the topography and emission sources in the NCP. Although the site is located in an area with rural/suburban development, it is impacted by anthropogenic emissions. The national capital, Beijing (population > 21 million), is located ~170 km in the northeast, and another megacity, Tianjin (population >15 million) is situated about 180 km to the east, while Shijiazhuang which is the capital and largest city of Hebei province (population >12 million) is 90 km to the southwest. In addition to these megacities, a prefecture-level city with population of ~11 million, Baoding, is 33 km to the northeast (Fig. 1b). The immediate surrounding area (i.e. within 5 km) of the sampling site is mostly covered by agricultural lands (Fig. 1c). The closest large local emission sources include a national highway and a provincial road, which are about 1-2 km away from the site. The major town area of Wangdu County is located ~5 km to the northwest while many densely spaced villages are sporadically spread around the area.





Dozens of coal-fired power stations are situated within a radius of 200 km. Among the nearest are Datang power station (capacity 650 MW) which is 27 km in the northeast and Dingzhou power station (capacity 2520 MW) which is 35 km in the southwest. Emission from the agriculture activities also have impacts on the site. The field study is in the harvesting season of winter wheat (Sun et al., 2007) and burning activities were frequently observed in the region, as indicated by the active fire hotspots obtained from FIRMS (MODIS C5, data available at https://earthdata.nasa.gov/firms) (see Fig. S1 in supplementary information (SI)). The less developed area of Taihang Mountains range (main peak = 2,882 m a.s.l) is located at 50-100 km in the north to west sector and the nearest coastline of Bohai Sea is ~200 km in the east.

## 2.2 Chemical ionization mass spectrometer

$ClNO_2$ and $N_2O_5$ were concurrently measured with a quadrupole chemical ionization mass spectrometer (THS Instruments, Atlanta). The principle and the calibration of the CIMS have been described in Wang et al. (2016). Briefly, iodide ions ($I^-$) were used as primary ions and the $N_2O_5$ and $ClNO_2$ were detected as ion clusters of $I(N_2O_5)^-$ and $I(ClNO_2)^-$ at 235 and 208 $m/z$, respectively. The CIMS measured $N_2O_5$ and $ClNO_2$ with a time resolution of ~7s. Data were later converted into 1 min averages for further analysis. During the Wangdu field study, the instrument background was determined by diverting the sampling flow through a filter fully-packed with activated carbons. Off-line calibrations of $N_2O_5$ and $ClNO_2$ were performed every day on the site, while standard addition of $N_2O_5$ into the ambient air was performed every 3 hours to monitor the sensitivity changes due to ambient conditions. More details on the calibration procedures can be found elsewhere (Wang et al., 2016). A corona discharge device (THS Instruments) was applied to generate $I^-$ from mixture of $CH_3I/N_2$ (0.3% v/v) at the beginning of the measurement (20 June – 26 June 2014) due to delay in shipment of a radioactive source. The large background signals from the corona discharge source (see Fig. S2) gave rise to detection limit of 16 pptv for $N_2O_5$ and 14 pptv for $ClNO_2$ ($3\sigma$, 1 min-averaged data). The corona discharge source was replaced by an alpha radioactive source, $^{210}Po$ (NRD, P-2031-2000) from 27 June 2014 until the end of the measurements. The detection limits for the latter period were improved to 7 pptv for $N_2O_5$ and 6 pptv for $ClNO_2$ ($3\sigma$, 1 min-averaged data).

The CIMS instrument was housed in a trailer. The sampling line was a 7.5-m long PFA-Teflon tubing (¼" O.D.). The inlet was set at ~2 m above the roof and ~10 m from ground level with a total sampling flow of ~11 standard liters per minute (SLPM). The inlet configuration was similar to a virtual impactor which is intended to remove large particles (e.g. Kercher et al., 2009; Kulkarni et al., 2011). Only ~4 SLPM from the total flow was diverted to the CIMS, ozone and $NO_x$ analyzer while the rest was dumped. The total residence time in the sampling system was less than a second. In order to minimize the effect of the particles deposited on the surface of the sampling inlet, the orifice, tubing and fittings were replaced and washed with ultrasonic every day (Wang et al., 2016). Examination of the measurement data did not show evidence of conversion of ambient $N_2O_5$ to $ClNO_2$ in the inlet. For instance, there were occasions when the $N_2O_5$ signal increased significantly with no



enhancement in $ClNO_2$, suggesting that the $ClNO_2$ was not produced in the inlet. The uncertainty of the measurement is estimated to be ±25% with a precision of 3%. The ambient measurements of $N_2O_5$ and $ClNO_2$ were available from 20 June to 9 July 2014.

## 2.3 Other measurements

The measurement techniques for trace gases and aerosols which are used to support the present analysis are summarized in Table 1. During the Wangdu study, most of the trace gases were simultaneously measured by different instruments/techniques. The agreement between these instruments/techniques and justification on the data set selections are discussed in another manuscript (Z. Tan et al., 2016, submitted to this issue of Atmospheric Chemistry and Physics). Basically, NO and $NO_2$ were measured by the chemiluminescence/photolytical conversion techniques, while total reactive nitrogen ($NO_y$) was determined by the chemiluminescence method with a molybdenum oxide (MoO) catalytic converter. $O_3$ was quantified by a UV absorption analyzer. Sulfur dioxide ($SO_2$) was measured by a pulsed UV fluorescence analyzer and carbon monoxide (CO) with an infrared photometer. $C_2$-$C_{10}$ hydrocarbons (NMHCs), formaldehyde (HCHO), and other oxygenated hydrocarbons (OVOCs) were measured with an online gas chromatograph (GC) equipped with a mass spectrometer and a flame ionization detector (FID), a Hantzsch fluorimetric monitor, and proton-transfer-reaction mass spectrometer (PTR-MS), respectively (Yuan et al., 2010, 2012; M. Wang et al., 2014). Methane was measured by cavity ring down spectroscopy technique (CRDs). Measurement of nitrous acid (HONO) was performed by a long-path absorption photometer (LOPAP) instrument (Li et al., 2014; Liu et al., 2016).

Particle mass concentrations ($PM_{2.5}$) were measured using a standard Tapered Element Oscillating Microbalances (TEOM). The ionic compositions of $PM_{2.5}$ were determined by a gas aerosol collector (GAC)-ion chromatography system (Dong et al., 2012). The dry-state particle number size distribution was determined by combining the data (Pfeifer et al., 2014) from a Mobility Particle Size Spectrometer (Dual TROPOS-type SMPS; Birmili et al., 1999; Wiedensohler et al., 2012) and an Aerodynamic Particle Size Spectrometer (TSI-type APS model 3321; Pfeifer et al., 2016) covering the size ranges from 4-800 nm (mobility particle diameter) and 0.8-10 µm (aerodynamic particle diameter), respectively. The ambient particle number size distributions as function of the relative humidity were calculated from a size-resolved kappa-Köhler function determined from real time measurement of a High Humidity Tandem Differential Mobility Analyzer (HHTDMA) (Hennig et al., 2005; Liu et al., 2014). Ambient particle surface area concentrations ($S_a$) were calculated based on the ambient particle number size distribution assuming spherical particles.

Meteorological parameters including wind direction, wind speed, relative humidity (RH), pressure and temperature were measured with an ultrasonic anemometer and a weather station on a 20-m height tower which was situated 30 m from



the trailers. Photolysis frequencies were determined from actinic flux densities measured by a spectroradiometer (Meteorologie Consult) (Bohn et al., 2008).

## 2.4 Meteorological and dispersion models

Weather Research and Forecasting model (WRF) was used for the simulation of meteorological fields during the study. Four nested domains were adopted for WRF simulations, covering whole China, northern China, North China Plain, and the surrounding area of the Wangdu site, with a grid size of 27, 9, 3 and 1 km, respectively. Other settings utilized in this study were the same as those described in Wang et al. (2016). The simulation results from the WRF were validated by using hourly

10   surface observation data obtained from China Meteorological Agency (CMA). WRF simulations generally reproduced the meteorology conditions in NCP during the campaign (refer to Table S1).

With the hourly WRF output, HYbrid Single-Particle Lagrangian Integrated Trajectory model (HYSPLIT) (Draxler et al., 2014) was adopted to investigate the history of air masses that arrived at the measurement site. The HYSPLIT model

15   was run in the dispersion mode for 12 hours backward in time, where 2500 particles were released at the sampling site and the hourly positions of these particles were tracked during this period. More detailed settings and descriptions of the HYSPLIT model can be found in Wang et al. (2016).

## 2.5 Chemical box model

In order to evaluate the contributions of $ClNO_2$ to daytime primary radical and $O_3$ production, an explicit observation-based chemical box model was utilized. The model was developed based on the latest version of Master Chemical Mechanism v3.3 (Jenkin et al., 2015) and was updated with a Cl chemistry module including 205 reactions of the inorganic mechanisms of Cl and VOCs degradations initiated by Cl (Xue et al, 2015).

The observation data of $ClNO_2$, HCl, HONO, $O_3$, NO, $NO_2$, $SO_2$, CO, $CH_4$, $C_2$-$C_{10}$ NMHCs, OVOCs (methanol, formaldehyde, acetone, acetaldehyde, acetic acid, MEK, MTBE), $H_2O$, temperature, pressure and aerosol surface area were averaged or interpolated and then constrained into the model every 10 min. The average concentration for each species and meteorological input are shown in Table S2. The photolysis frequency input of $NO_2$ ($j_{NO2}$), HONO ($j_{HONO}$), $O_3$ ($j_{O1D}$) and $ClNO_2$

30   ($j_{ClNO2}$) were determined from the field, while photolysis frequency of other related compounds were predicted following the function of solar zenith angle (Saunders et al., 2003) and were scaled according to the field measured $j_{NO2}$. The physical loss rate of the unmeasured species was set as the 6 h lifetime for the mixing height of 1000 m. The model was run for 24-hours period with the starting time set at 00:00 local time and was repeatedly run for 6 times to stabilize the unmeasured intermediate



species. The daytime output from the final run was used for further analysis of the primary radical production and $O_3$ production and loss processes.

## 3 Results and discussion

### 3.1 Overview of measurement results

Figure 2 depicts the temporal variations of $ClNO_2$, $N_2O_5$, related trace gases, $PM_{2.5}$ and selected meteorological parameters for the study period. The data gaps were caused by technical problems, calibrations or maintenances of the instruments which usually took place in the afternoon of each day. Elevated $ClNO_2$ was measured in all of the 13 nights with full CIMS measurements which show typical night-time concentrations larger than 350 pptv. The highest $ClNO_2$ was observed on 20-21 June with maximum mixing ratio of 2070 pptv. There were several nights when $ClNO_2$ mixing ratios were less than 200 pptv (e.g. on 24-25, 28-29 June, and 8-9 July). The observed $ClNO_2$ levels at Wangdu are comparable with previous measurements made in both coastal (e.g. Osthoff et al., 2008; Riedel et al., 2012; Mielke et al., 2013) and inland sites (e.g. Thornton et al., 2010; Phillips et al., 2012; Riedel et al., 2013). As for $N_2O_5$, low concentrations (<200 pptv) were observed in every night, implying fast loss of $N_2O_5$, except in the night of 28-29 June when mixing ratio of up to 430 pptv were observed in the air masses with low humidity (RH = ~40%) and NO (<2 ppbv).

The observation of elevated $ClNO_2$ is in-line with the expectation of ubiquitous $ClNO_2$ precursors like $NO_x$, $O_3$ and aerosols in the NCP environment. As shown in Figure 2, afternoon mixing ratios of $O_3$ exceeded 90 ppbv on a majority of days, with a maximum value of 146 ppbv, indicative of intense photochemical reactions during the study period. $NO_x$ mixing ratios were in the range of 10-80 ppbv, which reflects strong emissions of $NO_x$ in the region. Similarly, aerosol loading was quite high, with $PM_{2.5}$ mass concentration larger than 60 µg m$^{-3}$ on most of the days, with the highest value of 220 µg m$^{-3}$.

Figure 3 shows the 12-h backward particle dispersion trajectories with 08:00 local time (LT) as the starting time during 21 June – 9 July 2014. There were no significant changes in the origins of air masses for those trajectories arriving at 00:00 and 14:00 (Fig. S3 and S4). The study period can be meteorologically separated into three parts. The first part, 21-23 June, indicates air masses from megacities of Beijing and Tianjin (passing over Baoding) in the northeast. The highest $ClNO_2$ level was observed in this period. The second part begins at 24 June and ends 7 July with large majority of air masses originating from the southern sector and passing over a portion of urban areas of Shijiazhuang. The $ClNO_2$ mixing ratios were in the range of tens pptv to 1.2 ppbv. The final part is 8-9 July with air masses mostly from the less developed mountainous areas in the northwest sector, and the $ClNO_2$ concentrations were low. The entire field campaign was therefore dominated by air masses from southern regions, which is the typical summertime condition in the NCP.

### 3.2 Diurnal variations



Figure 4a illustrates the mean diurnal variation of $ClNO_2$ and relevant chemical data during the campaign. $ClNO_2$ exhibited a clear diurnal cycle with accumulation of $ClNO_2$ after sunset (~20:00) and reached a peak at ~08:00 in the morning. It then declined gradually to concentrations near the detection limit at noon. The average mixing ratios of $ClNO_2$ were up to 550 pptv. Its precursors, $N_2O_5$, only showed a small peak right after sunset with maximum average mixing ratio of 80 pptv, and remained at levels near the detection limit of the CIMS for the rest of the night. The $NO_y$, $NO_x$ and $S_a$ also showed a similar pattern as $ClNO_2$. They increased at sunset with average nighttime concentration of 29 ppbv, 21 ppbv and 1880 $\mu m^2$ $cm^{-3}$, respectively, and were at lowest levels in mid-day. The average night-time $NO_x$ to $NO_y$ ratio was 0.72. Diurnal variation of $O_3$ was anti-correlated with that for $NO_x$, with the former concentration rapidly decreasing as night falls.

The highest mixing ratio of $ClNO_2$ was observed on 20-21 June in the outflow of Tianjin megacity (see Fig. 3). We termed it as the megacity case in the remaining of the paper. The $ClNO_2$ mixing ratios in the megacity case were in the range of 110 to 2070 pptv, while $N_2O_5$ peaked at 170 pptv (Figure 4b). $NO_y$, $O_3$ and $S_a$ were generally at similar levels with the average condition, but the $NO_x$ was less abundant at this night compared to the campaign average, with a mean value of 16 ppbv. Smaller $NO_x/NO_y$ ratio of ~0.55 were found on this night, indicating more aged air masses being sampled.

### 3.3 Factors affecting ClNO₂ production

In this section, we examine the factors that may have caused the large difference of $ClNO_2$ levels in the megacity case and campaign average. Ambient $ClNO_2$ concentrations are affected by several factors including 1) production rate of $NO_3$ ($P(NO_3)$), 2) $N_2O_5$ reactivity (i.e. heterogeneous loss on aerosol surface, homogeneous reaction and dissociation to $NO_3$) and 3), production yield of $ClNO_2$ ($\phi$). The calculated nighttime $P(NO_3)$ through R1, $P(NO_3) = k_{NO2+O3}[NO_2][O_3]$, do not show much difference, with $1.7 \pm 0.6$ ppbv $h^{-1}$ in campaign average and $1.3 \pm 0.5$ ppbv $h^{-1}$ in the megacity case. Estimation of $\phi$ from the laboratory-parameterization with measured aerosol chloride content (Roberts et al., 2009; Bertram and Thornton, 2009) give a comparable $\phi$ of ~0.7 in the two cases.

The $N_2O_5$ reactivity was assessed with inverse $N_2O_5$ steady state lifetime analysis by using Eq (1) and (2) below (e.g. Platt et al., 1984; Brown et al., 2003, 2006, 2009, 2016).

$$\tau(N_2O_5)^{-1} = \frac{P(NO_3)}{[N_2O_5]} = \frac{k(NO_3)}{K_{eq}[NO_2]} + k(N_2O_5)_{het} + k_{Homo} \qquad (Eq\ 1)$$

$$k(NO_3) = k_{NO+NO3}[NO] + \sum_i k_i [VOC_i] \qquad (Eq\ 2)$$

The steady state inverse lifetimes of $N_2O_5$, $\tau(N_2O_5)^{-1}$, is the sum of the $N_2O_5$ loss rate through $NO_3$ (i.e. $k(NO_3)/K_{eq}[NO_2]$), $N_2O_5$ heterogeneous loss rate coefficient ($k(N_2O_5)_{het}$) and gas phase reaction of $N_2O_5$ with water vapor ($k_{Homo}$) (see Eq 1). The total reactivity can be obtained by the ratio of $P(NO_3)$ to the observed $N_2O_5$ mixing ratios (Brown et al., 2009). $K_{eq}$ is the



temperature-dependent equilibrium coefficient in R5, and the $k(NO_3)$ is the loss rate coefficient of $NO_3$ with NO and VOCs (see Eq2). Thus $k(N_2O_5)_{het}$ can be obtained by subtracting $k(NO_3)/K_{eq}[NO2]$ and $k_{Homo}$ from the determined $\tau(N_2O_5)^{-1}$. We only conduct analysis for the period between ~20:30 (0.5 h after sunset) until ~23:30 when there was no significant NO plumes (refer Figure 4), as interception of fresh emissions could lead to the failure of the $N_2O_5$ steady-state approximation in the air mass (e.g. Brown et al. 2003, 2011, 2016).

Figure 5a shows the averaged total $N_2O_5$ reactivity and fractions of $N_2O_5$ loss through $NO_3$, homogeneous and heterogeneous loss of $N_2O_5$ for both campaign average and the megacity case. The determined $\tau(N_2O_5)^{-1}$ is $1.3 \times 10^{-2}$ s$^{-1}$ for campaign average and $5.8 \times 10^{-3}$ s$^{-1}$ for the megacity case, suggesting that the average total loss rate coefficient of $N_2O_5$ is twice of that of the megacity case. However, the $N_2O_5$ reactivity is mainly dominated by loss via $NO_3$ (89%) for the campaign average, which is in-line with its relatively higher VOCs and NO background (Fig. S5). For the megacity case, although it has lower total $N_2O_5$ reactivity, the $k(N_2O_5)_{het}$ has about equal contribution with the loss via $NO_3$ and is about a factor of 2.4 faster than the campaign average.

The $ClNO_2$ production rate depends on the $N_2O_5$ heterogeneous loss rate coefficient, mixing ratios of $N_2O_5$ and $\phi$, which can be predicted with equation (3) when the loss of $ClNO_2$ is negligible during the night-time.

$$\frac{d[ClNO_2]}{dt} = k(N_2O_5)_{het}[N_2O_5]\phi \qquad (Eq\ 3)$$

As illustrated in Figure 5b, the predicted $ClNO_2$ production rate is a factor of 4 larger than the campaign average. The larger $ClNO_2$ production rate can be justified by the twice higher $k(N_2O_5)_{het}\phi$ and more abundant $N_2O_5$ (c.a. 2 times larger) in the megacity case that was due to the less $N_2O_5$ loss through conversion to $NO_3$ (as shown above). This result is consistent with the observed fourfold higher $ClNO_2$ mixing ratios in the megacity case compared to the average condition (c.f. Fig. 4), demonstrating that the faster heterogeneous $N_2O_5$ loss and smaller loss via $NO_3$ in the megacity case were the major reasons contributing to the larger $ClNO_2$ concentrations.

**3.4 Sustained ClNO₂ morning peaks**

A distinct feature of the $ClNO_2$ is the elevated concentrations sustained after sunrise. Figure 6 depicts the expanded view of the morning $ClNO_2$ peaks together with related chemical characteristics in the campaign average and megacity case. $ClNO_2$ concentration continued to increase after sunrise (at ~04:40) and persisted for 4 hours from sunrise for almost every day. The average mixing ratio of the morning $ClNO_2$ peak was 550 pptv with the megacity case reaching ppbv level. These results are different from the typical diurnal patterns of $ClNO_2$ observed at other places, which usually show a decline of $ClNO_2$ levels at sun rises (e.g. Osthoff et al., 2008; Thornton et al., 2010; Mielke et al., 2013; Tham et al., 2014). As outlined in the introduction,



morning peak of ClNO$_2$ was also observed in London (Bannan et al., 2015) and Texas (Faxon et al., 2015), but they were much smaller than the values at Wangdu.

### 3.4.1 Causes of ClNO$_2$ morning peaks

The ClNO$_2$ enhancement (ΔClNO$_2$) in the morning could be caused by in-situ ClNO$_2$ production and/or downward mixing of the ClNO$_2$ which has been produced in the residual layer (RL) over the night. We calculated the in-situ production of ClNO$_2$ (the area shaded in light grey in the Fig. 6) by using equation (4), which is similar to equation (3), with additional consideration of ClNO$_2$ loss via photolysis. Since the $k(N_2O_5)_{het}$ determined earlier (in section 3.3) are no longer applicable during the

10 daytime, the N$_2$O$_5$ heterogeneous loss rate here was estimated from equation (5), where γ is the N$_2$O$_5$ uptake coefficient and $c_{N2O5}$ is the mean molecular speed of N$_2$O$_5$.

$$\frac{d[ClNO_2]}{dt} = k(N_2O_5)_{het}[N_2O_5]\phi - j_{ClNO2}[ClNO_2] \tag{Eq 4}$$

$$k(N_2O_5)_{het} = \frac{1}{4}c_{N2O5}S_a\gamma \tag{Eq 5}$$

We used a γ of 0.03 and unity ϕ of 1.0 in the calculations. These numbers are considered as upper end values based on previous

field studies (Brown et al., 2006; Bertram et al., 2009; Reidel et al., 2013). As shown in the lower panel of Figure 6, the calculated ΔClNO$_2$ with γϕ = 0.03 cannot reproduce the observed increases in ClNO$_2$. Larger γϕ of 0.06 - 0.09 would be needed, but such large uptake coefficients and yields are not supported by the currently available data in the literature. Therefore, we think that in-situ ClNO$_2$ production is not the main reason for the ClNO$_2$ morning peak.

Meteorological and chemical data point to the entrainment of ClNO$_2$ rich air aloft after sun rise as the cause of the ClNO$_2$ morning peaks. Figure 7 shows the fractions of air arriving at the measurements site from various altitudes at different time of day based on the simulations of WRF-HYSPLIT. Vertical mixing was limited prior to sunrise (~04:00) as most of the air masses were confined to ground level (< 200 m above ground level, a.g.l.). Shortly after sunrise (~05:00), contributions of air masses from the higher levels began to increase after the break-up of nocturnal boundary layer (NBL). As time advanced,

larger fraction of higher-level air masses impacted the surface site. Chemical data is consistent with the meteorological analysis. As shown in Figure 6, the SO$_2$/NO$_y$ ratios in both cases increased up to 0.6-0.8 after sunrise, indicative of the impact of plumes from coal-fired facilities like power plants. The power-plant plumes from elevated stacks typically reside above the nocturnal boundary layer (NBL) due to poor mixing at night. Coal-fired power plants emit large amount of NO$_x$ and Cl$^-$ containing aerosols, in addition to SO$_2$ (McCulloch et al., 1999; Zhao et al., 2008). Together with high O$_3$ produced in the

preceding daytime and aerosol loadings, significant production of ClNO$_2$ above the NBL are expected and indeed have been observed in previous field studies (Wagner et al., 2012; Young et al., 2012; Riedel et al., 2013). In the present study, the ClNO$_2$ precursors like Cl$^-$ aerosol and P(NO$_3$) and a co-product of chlorine activation, nitrate (NO$_3^-$) aerosol also showed significant enhancement in the early morning hours (see Fig. 6).



### 3.4.2 Estimation of ClNO₂ concentrations in the residual layer

We estimate the amount of the $ClNO_2$ that would exist in the residual layer to maintain the observed $ClNO_2$ at ground level. Here we use a simplified one dimensional (1-D) model to illustrate the mixing process. This model contains two layers of air before sunrise; NBL and RL, with $ClNO_2$ concentrations of $C_n$ and $C_r$, respectively (see Fig. S6). We assume no mixing of air masses (and $ClNO_2$) between the two layers. After sunrise, the two layers are efficiently mixed, yielding a constant concentration of $ClNO_2$ ($C_p$). The height of the daytime planetary boundary layer (PBL) and NBL were calculated by the WRF model. The difference in the heights of PBL and NBL is the depth of the RL. $C_n$ and $C_p$ are the observed mixing ratios before (at 05:00) and after (at 08:00) sunrise, respectively. The concentration in RL layer before sunrise can be estimated by the mass balance approach taking consideration of loss of $ClNO_2$ from photolysis between 05:00 and 08:00 (Eq (6)).

$$C_p \times H_p = (C_n \exp^{(-jClNO2t)} \times H_n) + (C_r \exp^{(-jClNO2t)} \times H_r) \qquad \text{(Eq 6)}$$

Where $j_{ClNO2}$ is photolysis rate of $ClNO_2$ and $t$ is time.

For the campaign average, WRF calculated boundary layer height is 30 m and 325 m (a.g.l.) at 5:00 and 8:00, respectively (refer to Fig. S7). This gives a mixing ratio of $ClNO_2$ in RL of 1.7 ppbv. For the megacity case with boundary layer height of 72 m at 05:00 and 610 m at 08:00, the $ClNO_2$ in RL by sunrise would be 4.0 ppbv. These values are within the range of aircraft and tower measurements in RL in the US (Wagner et al., 2012; Young et al., 2012; Riedel et al., 2013) and are comparable to the highest $ClNO_2$ observed at a mountain site in southern China (Wang et al., 2016). This result suggest that elevated $ClNO_2$ may always present in the residual layer of this region.

### 3.5 Sources of chloride aerosols

The elevated $ClNO_2$ at Wangdu site requires sufficient amount of chloride aerosols to support its production. Abundance of fine $Cl^-$ aerosols were frequently observed during night hours (20:00-09:30) with a mean concentration of 1.6 µg m$^{-3}$ and maximum of 6.8 µg m$^{-3}$ (Fig. 8). The $Cl^-$ concentrations in our study are comparable to those previously observed in the NCP (Sun et al., 2006; Huang et al., 2014; Sun et al., 2015). As can be seen in Figure 3, back-trajectories at Wangdu indicated the air was mainly of continental origins with limited direct influences from the oceans. Chemical data also provide evidence for non-oceanic $Cl^-$ sources. The $Cl^-$ aerosol showed good correlation ($r$ >0.75) with $SO_2$ in 11 out of 16 nights, including 4 nights with concurrent good correlation ($r$ >0.75) with a biomass burning tracer of acetonitrile ($CH_3CN$) (see Table 2). These results suggest that coal-fired power plants are a dominant source of chloride in the region with additional contributions from biomass burning. Significant chlorine content (260 mg kg$^{-1}$) has been found in the coal used in China (Zhang et al., 2012). Under high temperature and oxygen free conditions, combustion of coal can release up to 97% of the chlorine in the coal in the form of HCl gas (Gibb, 1983) which can then be transformed into aerosol phase ($Cl^-$) through neutralization reactions in the ambient



air. The contribution of biomass burning at Wangdu can also be seen in the active fires data on the nights with good correlation between $Cl^-$ aerosol and $CH_3CN$. Figure 9 shows an example of burning activities mostly in south of Wangdu on 28-29 June. Li et al (2007) measured composition of smoke from burning of wheat straw and maize stover harvested in NCP and found 13.8% and 23.0% of $Cl^-$ in the $PM_{2.5}$ mass loading, respectively. Recent field measurement of biomass burning plumes during

the harvesting period in China also indicated drastic increase in the $Cl^-$ concentration ($>20$ µg m$^{-3}$) (Li et al., 2014).

### 3.6 The impact of ClNO$_2$ on primary radical and ozone production

This section examines the contributions of $ClNO_2$ to the primary $RO_x$ ($OH+HO_2+RO_2$) radical and in-situ ozone production at

10 Wangdu using the observation-constrained box model described in Section 2.5. The analysis focuses on campaign average condition and the megacity case. The mean concentrations of trace gases and other parameters that serve as inputs are shown in Table S2.

Figure 10a illustrates the Cl production rate derived from the photolysis of $ClNO_2$ and without $ClNO_2$ (from photolysis

of $Cl_2$ and HCl+OH). It shows that photolysis of $ClNO_2$ was the predominant source of Cl in Wangdu. The production of Cl was efficient in the morning (from sunrise to ~11:00) and reached maximum at ~08:00 corresponding to the peak concentration of $ClNO_2$. The Cl production rate was up to 0.24 ppbv h$^{-1}$ for average condition, and up to 1.12 ppbv h$^{-1}$ for the megacity case. Figure 10b depicts the primary daytime $RO_x$ production ($P(RO_x)$) from sources including photolysis of $ClNO_2$, OVOCs (excluded HCHO), HCHO, HONO and $O_3$ ($O^1D+H_2O$), $O_3$+VOCs, and $NO_3$ oxidations. Similar to other previous studies, the

primary daytime $P(RO_x)$ is dominated by sources from photolysis of HCHO, OVOCs and HONO (Kanaya et al., 2009; Liu et al., 2012; Lu et al., 2013). The photolysis of $ClNO_2$ is particularly important during the morning hours. During 08:00 - 08:30, photolysis of $ClNO_2$ contributed to 10% of the $P(RO_x)$ on average condition and with a much larger contribution of 30% in the megacity case. These results highlight the importance of $ClNO_2$ as a significant source of $RO_x$ radicals in this region. Reader is referred to Z. Tan et al., (2016, submitted to this issue of Atmospheric Chemistry and Physics) for more extensive

analysis on the $RO_x$ chemistry at Wangdu.

The effect of $ClNO_2$ photolysis on in-situ ozone production is also relevant. Figure 11 shows the net ozone production rates ($P(O_3)$) during daytime (from 05:00 to 18:00) and the difference of integrated total ozone production simulated with $ClNO_2$ and without $ClNO_2$ input. The $O_3$ production rates were enhanced throughout the day due to the $ClNO_2$ effect, especially

during the morning hours. The increase of net $P(O_3)$ for campaign average reached 0.9 ppbv h$^{-1}$ or 17% during the morning. For the megacity outflow, much higher increases in $P(O_3)$ can be seen in the entire morning, with a maximum of 3.3 ppbv h$^{-1}$ (or 76% increase) at ~08:00. Integrating over the entire daytime period, the increase of total ozone production was 4.3 ppbv (3%) and 11 ppbv (13%) for average condition and the megacity case, respectively. Although not directly comparable, these



values are generally within the range of net ozone production increase caused by $ClNO_2$ in previous studies in Houston (Osthoff et al., 2008), Los Angeles (Riedel et al., 2014), and southern China (Xue et al., 2015; Wang et al., 2016).

We notice a large difference in the impact of $ClNO_2$ on ozone production between campaign average and the megacity case. This can be explained by their $ClNO_2$ and VOCs characteristics. First, for campaign average, $ClNO_2$ mixing ratio was much lower than that of the megacity case (see section 3.3). The smaller $ClNO_2$ concentrations in turn would produce less chlorine radical in the daytime, reducing the production of $RO_x$ which ultimately decreases the $O_3$ production. A simulation test by only reducing the $ClNO_2$ mixing ratios in the megacity case by a factor of 2.8 (that is, to the same levels of campaign average) showed a sharp drop in the increase of the ozone production from 13% to 6% (Fig. S8a), confirming the importance of $ClNO_2$ mixing ratios in driving the ozone enhancement. Second, the higher OVOC to NMHC fraction in campaign average (see Fig. S9 for the VOCs mix) provided a larger pool of $RO_x$, and the radical propagation would amplify the OH through efficient radical recycling (Liu et al., 2012), dampening the effect of chlorine radical. Another test by only increasing the $ClNO_2$ mixing ratios in campaign average condition to the same value of the megacity case indicated a relatively smaller increase of the ozone production (9%) (see Fig. S8b) compared to the increased percentage of $O_3$ production in the megacity case (13%).

The much higher OVOCs mixing ratios to NMHCs in campaign average are likely influenced by the biomass burning activities which are more intense in the regions south of Wangdu (refer to Fig. S1) and the biomass burning-influenced plumes can be transported to Wangdu by prevailing southerly winds as shown by the trajectories (Fig. 3). During our field measurement, we indeed observed significant increase (up to 170 ppbv) of OVOCs in a fresh biomass burning plume on the midnight of 15 June 2014 (Fig. S10). High content of OVOCs from biomass burning has been previously reported at a mountain–site of the NCP (Inomata et al., 2010). Significant emissions of OVOCs from the burning of Chinese crop residues were also reported in a recent laboratory study (Inomata et al., 2015).

We hence conclude that higher concentrations of $ClNO_2$ and less abundant of VOCs (i.e. smaller OVOC to NMHC fraction) in the megacity case resulted in a relatively higher impact of Cl chemistry on the ozone formation. Although we only measured very high $ClNO_2$ in one event of megacity outflow, similar cases occurred in the beginning of the study period during which $NO_x$, $NO_y$, $S_a$ and VOCs data were available (but not $ClNO_2$). An examination of the VOCs mix and other chemical compounds confirmed the less abundance of OVOCs but has similar chemical characteristics in these cases (see Fig. S11 for an example of chemical characteristics on 23 and 27 June 2014). Previous studies have shown less abundance of OVOCs relative to NMHCs in and downwind of urban Beijing (Liu et al., 2009; Xu et al., 2011). Thus, we suggest that the effect of $ClNO_2$ on ozone enhancement may be more important in air masses dominated by urban/power plants emissions than those by biomass burning.



## 4. Summary and conclusions

This first ClNO$_2$ measurement in northern China unambiguously documented the presence of elevated ClNO$_2$ in this high polluted region. Highest ClNO$_2$ mixing ratio (2070 pptv, 1 min average) was observed in urban outflow on 20-21 June 2014.
The air mass was characterized by faster (by a factor of 2.4) N$_2$O$_5$ heterogeneous loss as well as larger ClNO$_2$ production rate (by a factor of 4) compared to campaign average condition. The peak concentrations of ClNO$_2$ often occurred 4 hours after sun rise. Downward mixing of ClNO$_2$ rich air in the residual layer is believed to be the cause of these morning peaks, and the mixing ratios of ClNO$_2$ in RL are estimated in the range of 1.7-4.0 ppbv. These values are supported by our mountain-top measurement of ClNO$_2$ in Hong Kong with up to 4.7 ppbv in well processed urban/industrial plumes (Wang et al., 2016). The Wangdu result implies strong productions of ClNO$_2$ in the residual layers over the polluted regions of northern China. We also present evidence for existence of fine aerosol chloride from non-oceanic sources like coal-fired power plants and burning of crop residues, suggesting widespread effects of ClNO$_2$ on oxidative capacity and production of secondary pollutants in this region. Our model calculations suggest larger impact of ClNO$_2$ on primary radical productions and ozone enhancement in urban/power plant emission dominated air masses compared to biomass burning due to higher levels of VOCs (also larger OVOC to NMHC ratios) and relatively lower ClNO$_2$ in the latter reducing the effect of ClNO$_2$. More studies in non-biomass burning seasons and in areas adjacent to megacities/power plants are needed to examine the production of ClNO$_2$ and its effect on the regional photochemistry. Vertical profile measurements of ClNO$_2$ and related species would also be highly desirable.

**Acknowledgement.** The authors thank Steven Poon, Zha QiaoZhi, Xu Zheng and Wang Hao for the logistics support, to Zhang Li, Liu Xiaoxi, and Steve Sjostedt for helps in the modeling and instrumentations, to the CARE-Beijing 2014 team for their contributions in obtaining the data during field campaign. We are grateful to China Meteorological Agency for providing observational meteorological data and to NOAA Air Resources Laboratory for providing the HYSPLIT model. This work was funded by the National Natural Science Foundation of China (41275123) and PolyU Project of Strategic Importance (1-ZE13). The Peking University team acknowledges support from National Natural Science Foundation of China (21190052) and Strategic Priority Research Program of the Chinese Academy of Sciences (XDB05010500). The Leibniz Institute for Tropospheric Research team acknowledge financial funding by Sino German Science center (No. GZ663).

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



**Table 1. Measurement method for trace gases and aerosols.**

| Species | Measurement Techniques | Detection Limits | Time resolution |
|---|---|---|---|
| $ClNO_2$, $N_2O_5$ | CIMS | 6-7 pptv | 1 min |
| $O_3$ | UV photometry | 0.5 ppbv | 1 min |
| NO | Chemiluminescence | 0.06 ppbv | 3 min |
| $NO_2$ | Photolytical converter & Chemiluminescence | 0.3 ppbv | 1 min |
| $NO_y$ | MoO catalytic converter & Chemiluminescence | <0.1ppbv | 1 min |
| $CH_4$ | CRDs | 0.1 ppmv | 1 min |
| $SO_2$ | Pulsed-UV fluorescence | 0.1 ppbv | 1 min |
| CO | IR photometry | 4 ppbv | 1 min |
| HONO | LOPAP | 7 pptv | 0.5 min |
| HCl | GAC-IC | 59 pptv | 30 min |
| NMHCs | GC-FID/MS | 20 – 300 pptv | 60 min |
| OVOCs | PTR-MS | 10 - 50 pptv | 5 min |
| Formaldehyde | Hantzsch (wet chemical fluorimetric) | 25 pptv | 1 min |
| $PM_{2.5}$ | TEOM | 2 $\mu g/m^3$ | 1 min |
| Aerosol ionic compositions | GAC-IC | 0.01-0.16 $\mu g/m^3$ | 30 min |



**Table 2. Correlations of chloride with power plant and biomass burning indicators (from 20:00-09:30 LT).**

| Duration (20:00-09:30) | [Cl⁻] | Power plant indicator (SO$_2$) | | | Biomass burning indicator (CH$_3$CN) | | |
|---|---|---|---|---|---|---|---|
| | Mean (µg m$^{-3}$) | Mean (ppbv) | Slope | Correlation (r) | Mean (ppbv) | Slope | Correlation (r) |
| 20-21 June | 1.450 | 5.74 | 0.092 | 0.858 | 0.35 | 2.20 | 0.330 |
| 21-22 June | 2.792 | 3.87 | 0.267 | 0.963 | n/a | n/a | n/a |
| 23-24 June | 1.729 | 5.65 | 0.234 | 0.946 | 0.43 | 0.10 | 0.003 |
| 24-25 June | 0.760 | 4.88 | 0.235 | 0.852 | 0.39 | 36.96 | 0.547 |
| 25-26 June | 1.988 | 2.79 | 0.281 | 0.833 | 0.56 | 11.27 | 0.893 |
| 27-28 June | 0.335 | 6.67 | 0.008 | 0.213 | 0.39 | -1.33 | 0.577 |
| 28-29 June | 0.915 | 7.44 | 0.083 | 0.831 | 0.55 | 2.77 | 0.830 |
| 29-30 June | 1.429 | 22.67 | 0.048 | 0.719 | 0.48 | 4.96 | 0.442 |
| 30 June – 1 July | 1.283 | 10.43 | 0.094 | 0.781 | 0.45 | 6.01 | 0.268 |
| 01-02 July | 0.762 | 2.53 | 0.089 | 0.630 | 0.32 | 4.76 | 0.448 |
| 02-03 July | 2.187 | 3.84 | 0.195 | 0.762 | 0.43 | 10.68 | 0.548 |
| 03-04 July | 2.636 | 3.69 | 0.314 | 0.613 | 0.39 | 12.76 | 0.566 |
| 04-05 July | 2.158 | 2.55 | 0.365 | 0.922 | 0.35 | 30.09 | 0.924 |
| 05-06 July | 2.857 | 7.29 | 0.112 | 0.776 | 0.44 | 12.86 | 0.763 |
| 06-07 July | 1.720 | 6.04 | 0.113 | 0.853 | 0.46 | 9.84 | 0.502 |
| 07-08 July | 1.939 | 8.70 | 0.084 | 0.607 | 0.49 | 2.14 | 0.157 |







**Figure 1. Google map showing a) the location of northern China (white dash-box) and Wangdu site (blue triangle), b) an expanded view of northern China plain with the topography, major cities and the locations of major coal-fired power plants (yellow cross) in the region, c) nearby environment of Wangdu site.**





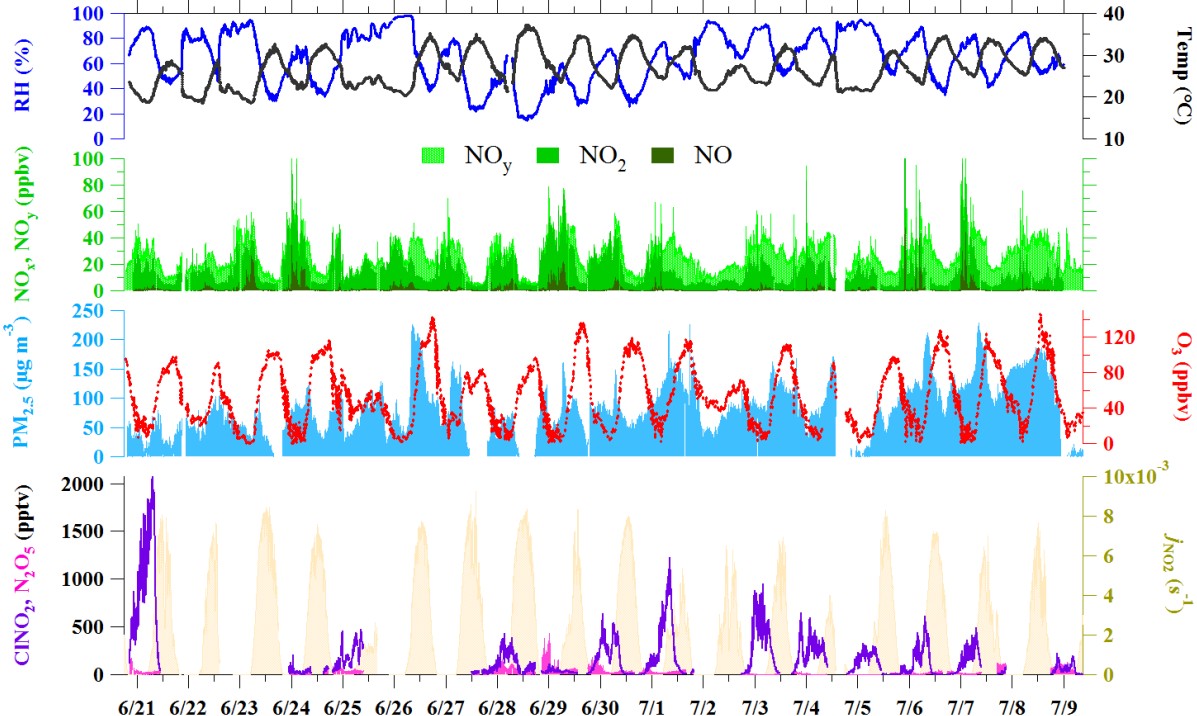

**Figure 2. Time series of ClNO₂, N₂O₅ together with related species and meteorological data. Data are at 1-minute time resolution, except for NOₓ data which are 5-minute averages.**





**Figure 3. 12-hour history of air masses arriving at the measurement site at 08:00. Yellow cross represents major coal-fired power plants in the region.**



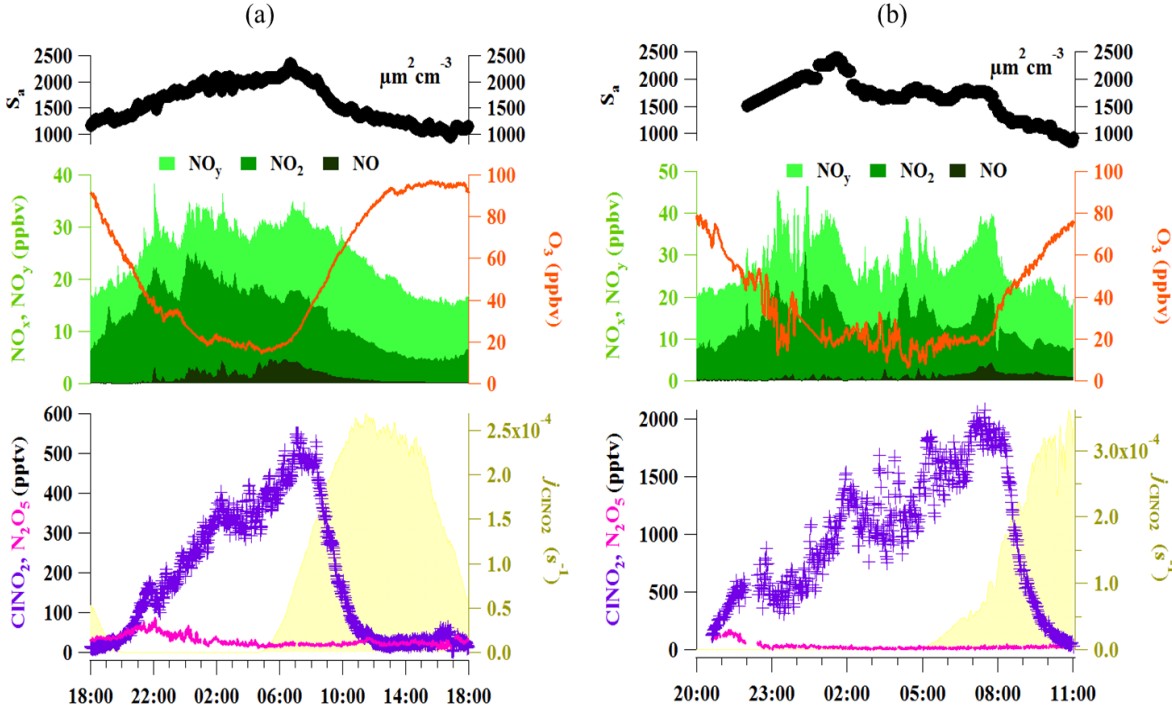

**Figure 4. Diurnal variations of ClNO₂, N₂O₅, NOₓ, NOᵧ, O₃ and particle surface area for a) campaign average (from 20 June to 9 July 2014 when ClNO₂ data is available) and b) the highest ClNO₂ case on 20-21 June 2014 (the megacity case).**



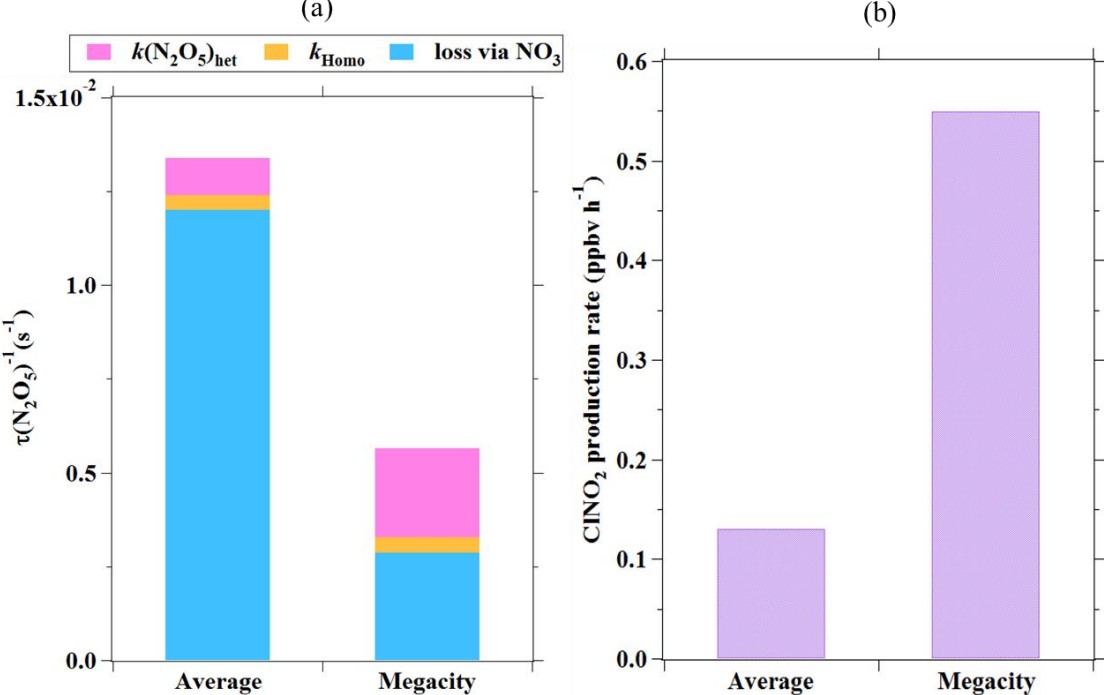

**Figure 5. a)** The fractions of $N_2O_5$ loss rate coefficient through $NO_3$, homogenous and heterogeneous reaction of $N_2O_5$ for campaign average and the megacity case; **b)** the predicted $ClNO_2$ production rate in the megacity case shows to be fourfold larger than campaign average.



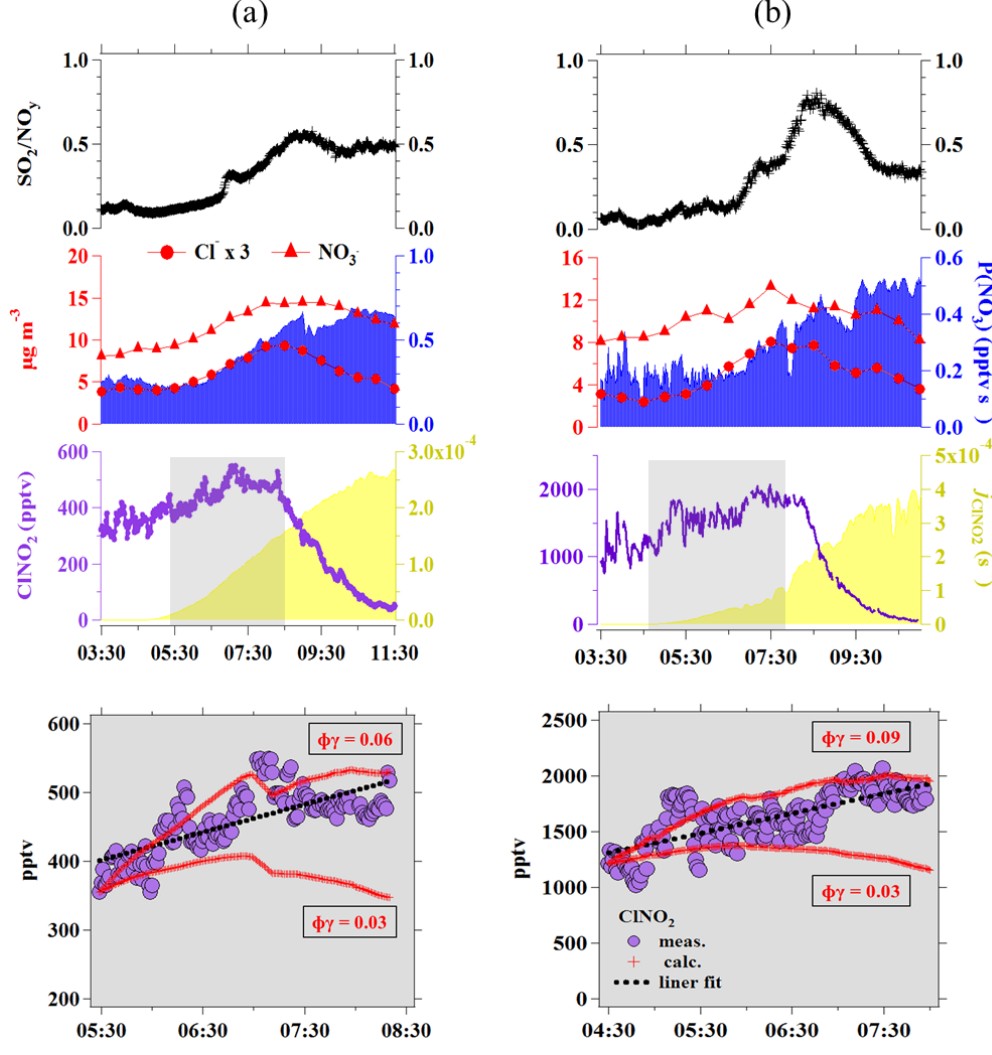

**Figure 6. Expanded view of the sustained morning ClNO₂ peaks for a) campaign average and b) the megacity case. The upper panel shows some relevant chemical information including SO₂/NOᵧ, Cl⁻, NO₃⁻, and P(NO₃). The increase of ClNO₂ in the shaded area (light grey) are compared to the calculated in-situ production of ClNO₂ (lower panel).**





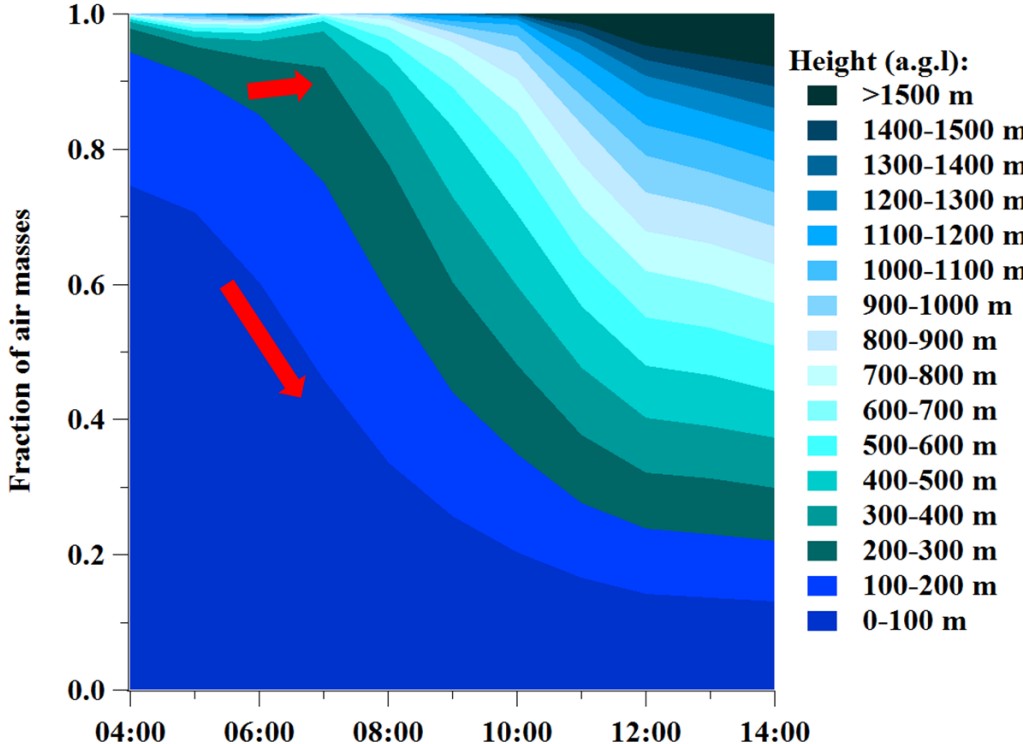

**Figure 7. Fractions of air masses arriving at the measurement site at different time of the day (average condition). It is noted that these fractions were derived from the 1 h backward-in-time HYSPLIT simulation results. Red arrows show the decreased contribution of lower layer (<200m) and increased contribution from upper layer (>200m) air masses shortly after sunrise.**



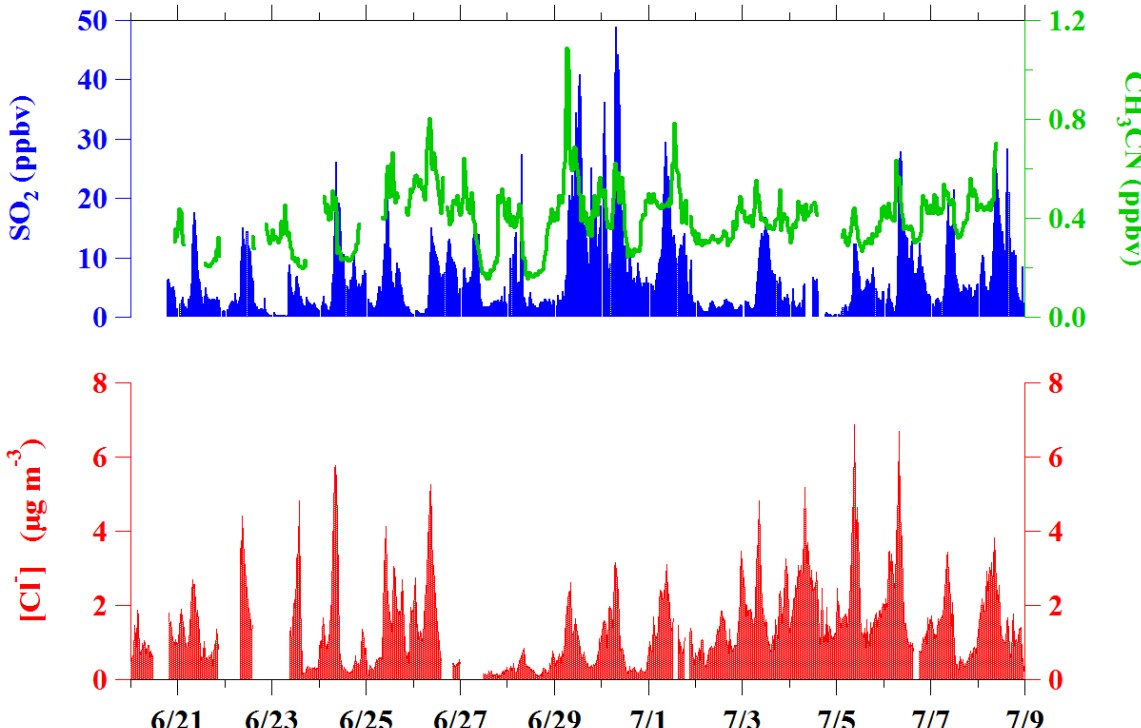

**Figure 8. Time series of Cl⁻ aerosol, SO₂ (a coal-fired power plant indicator) and CH₃CN (a biomass burning indicator) from 20 June to 9 July 2014.**





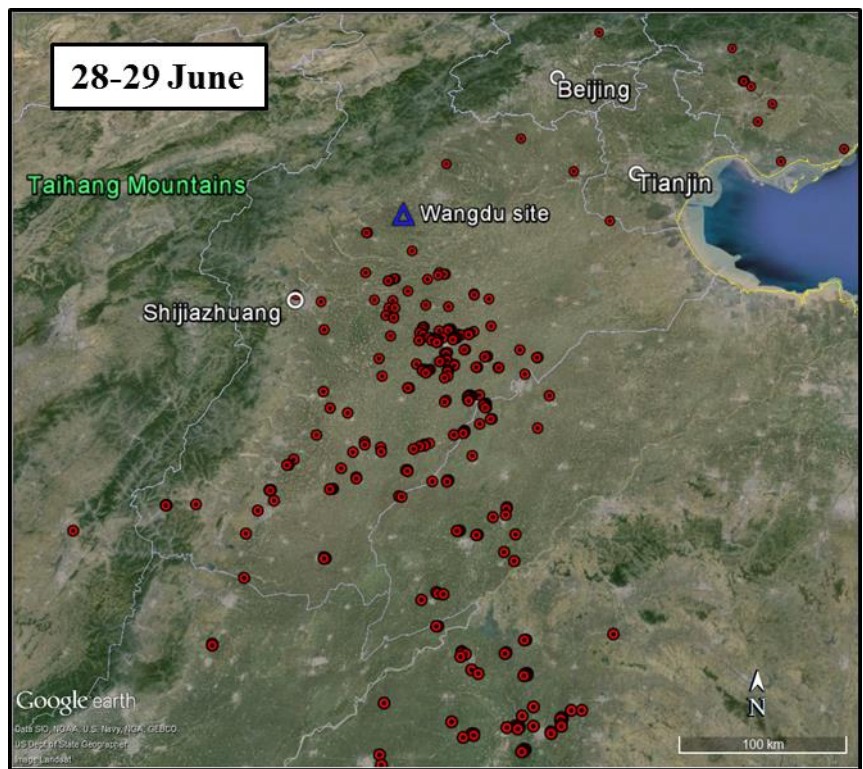

**Figure 9. Biomass burning activities from the active fire hotspots data (red dots) on 28-29 June (Data available at https://earthdata.nasa.gov/firms).**





**Figure 10.** a) Cl production rates with ClNO₂ and without ClNO₂ input; b) primary ROₓ radical production rates from different sources at Wangdu in the daytime. Pie charts represent the contributions of ClNO₂ to the primary ROₓ radical production in the morning time (average value for 08:00-08:30).





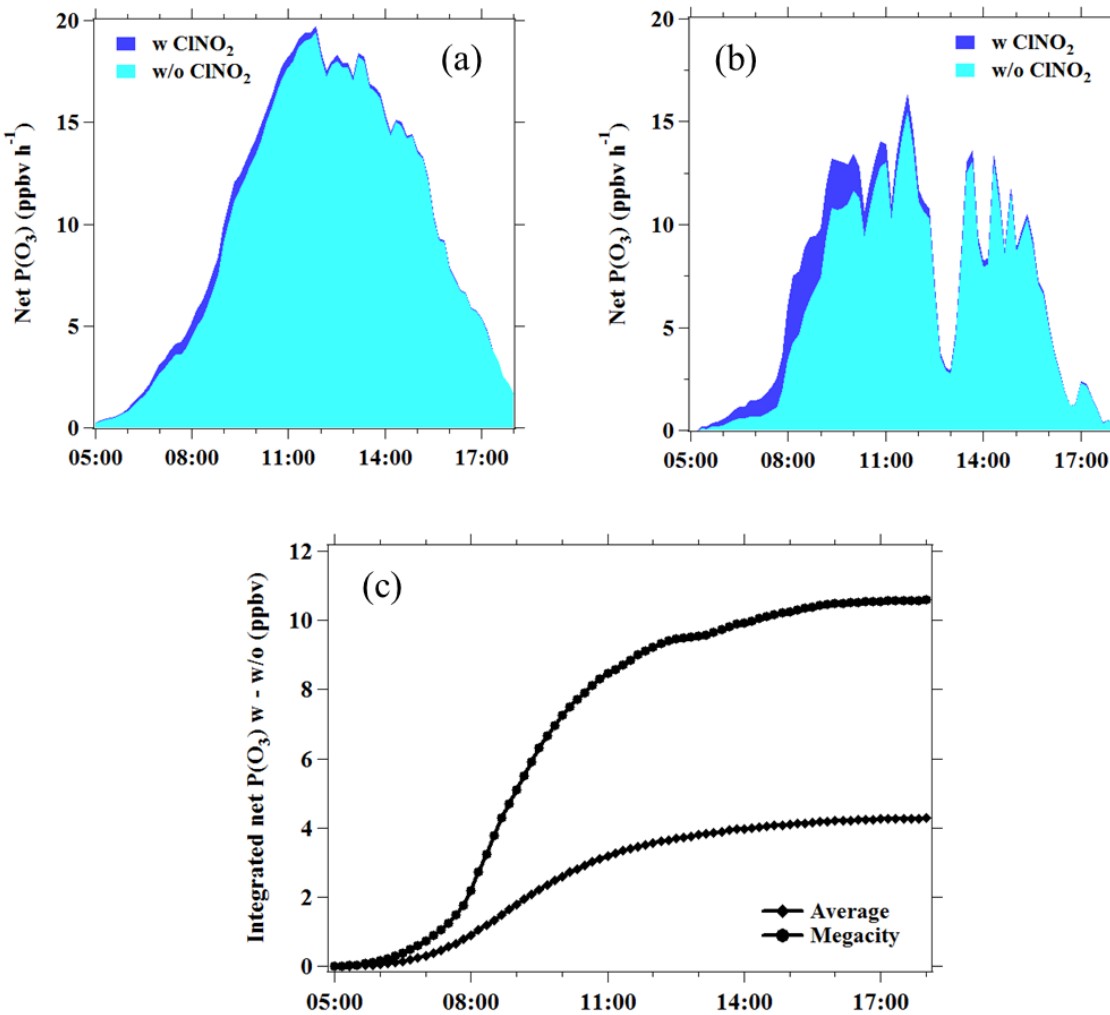

**Figure 11. Net ozone production rates for a) average, b) the megacity case and c) the difference of integrated net ozone production rate between the simulation with and without the ClNO₂ input.**

