# Peer review of "Significant concentrations of nitryl chloride sustained in the morning: Investigations of the causes and impacts on ozone production in a polluted region of northern China"

_Atmospheric Chemistry and Physics, 2016_

## Referee Comment (RC1) · Anonymous Referee #1 · 29 Jun 2016

Tham et al. present a comprehensive set of measurements and analysis focusing on ClNO2 formation at a ground site in Northern China. I thought this was a well written manuscript which should be published after my suggestions below have been considered.

Minor issues.

pg 2 line 12. (R5) please use the proper chemical symbol for a reversible reaction (the symbol used denotes resonance)

pg 5 - section 2.2. What were the response factors for N2O5 and ClNO2 at m/z 235

and 208 when the Corona discharge and the 210Po were used?

It may be worthwhile to add more detail about the calibration here, and add a figure of an example calibration sequence to the supplemental. The Wang et al. (2016) describes results from a different study, where there was a CRDS N2O5 instrument.

Was the humidity in the CIMS inlet controlled?

I am concerned about the measurement of N2O5 using m/z 235. Can you comment on potential interferences arising from clustering of iodide with organic acids?

Is m/z 210 consistent with the relative isotopic abundance of 37Cl?

pg 7, line 30 - the ClNO2 cross-sections were remeasured in 2008 by Ghosh et al. (JPC A 116, 6003 (2012)). Please indicate which cross-sections were used in this work.

pg 9 line 29. The homogeneous hydrolysis rate by Wahner et al. is likely incorrect (see, e.g., Brown et al., Science, 2006). Consider omitting it.

pg 11 line 15. His last name is spelled Riedel.

pg 12, equation (6). There may also be "loss" of ClNO2 due to entrainment upwards from the residual layer (not just downward mixing). Hence, the levels in the residual layer could be higher than calculated here.

pg 12 line 19 "This result suggest that elevated ClNO2 may always present in the residual layer of this region." One cannot logically conclude from some observations to "always" as there may be the odd exception. Suggest rephrasing to "frequently" or similar.

pg 15 line 27- many references are incomplete (missing doi, volumes, page numbers, etc.).

pg 22 (Table 1). Please state the uncertainties for each of the measurements.

General. There are a few minor grammatical errors scattered throughout the document. I would suggest asking a native English speaker to read through the manuscript a couple of times and make corrections where warranted.

---

## Referee Comment (RC2) · Anonymous Referee #2 · 30 Jun 2016

Summary: Tham et al. present a novel set of measurements of ClNO2, N2O5, along-side supporting observations of select trace gases and aerosol. The observations provide new insight on under sampled regions of the atmosphere, particularly with respect to molecules that are recently emerging as being important for atmospheric oxidation. The analysis follows prior work in this area conducted in the US and Europe and is well founded in the observations. I have a few select comments that should be addressed prior to publication. There are also a series of grammatical errors that should be addressed before the paper makes its way to publication in ACP.

[Figure]

Specific Comments:

1) Page 3 lines 1-5: Perhaps discuss in terms of the ClNO2 photolysis lifetime instead of concentrations following sunrise.

2) Page 5 line 10: The use of I- ion chemistry in extremely polluted regions such as this is limited and it is perhaps likely that other atmospheric compounds contribute to the observed signals at 235 and 208 m/z. A few quality control questions: 1) What is the 208/210 ratio for ClNO2 detection, is this consistent with the natural abundance of Cl isotopes? and 2) is there any signal intensity during the daytime (midday / late afternoon) at 235 and 208/210 m/z that would indicate a contribution from other molecules at these masses?

3) Page 7 line 31: What is meant by the "physical loss rate of the unmeasured species was set as the 6 h lifetime for the mixing height of 1000 m." Is deposition included in these models?

4) Page 10 line 6: Converting the computed lifetimes to reactive uptake coefficients based on measured Sa would be a helpful addition as the community is well calibrated to that language. It would also be helpful to include specific values for the ClNO2 yields that best fit the observations.

5) Page 11 line 10: What is the accuracy in the measured surface area? Is the surface area reported here dry or wet? If you need a factor of three change in gamma(N2O5) to match the data, is that within the uncertainty in Sa? Especially given that a growth factor may be needed to convert the measured dry to the relevant ambient Sa.

6) Page 12 line 15: The calculation of RL ClNO2 is very sensitive to the boundary layer height at 5 and 8AM. Are there measurements of this height? Also, what is the accuracy in the WRF calculated nocturnal boundary layer height? It is hard to imagine this is accurate to the values quoted here (50 and 72m).

---

## Referee Comment (RC3) · Anonymous Referee #3 · 29 Jul 2016

Review of Tham et al., ACP-2016-439

This paper describes measurements of nitryl chloride and associated species at a site in the North China Plain (NCP), and presents model estimates of the impact of this active chlorine compound on ozone formation in that environment. The measurements are very interesting and the associated analysis makes some important points about $ClNO_2$ in the residual boundary layer of the polluted NCP. The presentation of the work is quite well done, it is concise and well organized and the important aspects are well explained. There are only a few issues for the authors to address to make this paper acceptable for publication.

General Comments
In general the English in the paper is quite good, however there are a number of instances disagreements between the noun and the verb (e.g. singular when it should be plural, etc.). The authors briefly mention measurements of gas-phase HCl, but since this is an important fraction of the chloride available for activation, it deserves more details. Also, the morning time source of Cl atoms will have a corresponding source of HCl, as most Cl + VOC reactions produce HCl.

Specific Comments

Page 2, Line 31: While $ClNO_2$ is not as well studied as $N_2O_5$, there are loss mechanisms for $ClNO_2$ at night. Kim et al., [2014] show that $ClNO_2$ can be deposited on water surfaces. Roberts et al., [2008] showed that $ClNO_2$ can be taken up on low pH surfaces (and will make $Cl_2$).  It is fair to say that because of its low aqueous solubility [Sander, 2015], $ClNO_2$ losses are likely much slower than $N_2O_5$, and to a first approximation can probably be neglected.
Page 4., Lines 8&9. When you say "tropospheric ozone" that implies a broad scale, really you are talking about ozone in the Beijing urban area.
Page 5, Line 12. Did you see any evidence of $Cl_2$, at the mass of the cluster ion $I(Cl_2)^-$ ?
Page 7, Line 28, "constrained into" is the wrong expression, a model can be 'constrained by' observations.
Page 14, Line 29, "less' should be 'lesser'.
Figure 2. The yellow color is hard to see.

References

Kim, M. J., Farmer, D. K., and Bertram, T. H.: A controlling role for the air−sea interface in the chemical processing of reactive nitrogen in the coastal marine boundary layer, Proc. Natl. Acad. Sci., 10.1073/pnas.1318694111, 2014.

Roberts, J. M., Osthoff, H. D., Brown, S. S., and Ravishankara, A. R.: $N_2O_5$ oxidizes  chloride to $Cl_2$ in acidic atmospheric aerosol, Science, 321, 1059., 2008.

Sander, R.: Compilation of Henry's law constants (version 4.0) for water as solvent, Atmos. Chem. Phys., 15, 4399-4981, 10.5194/acp-15-4399-2015, 2015.

---

## Author Comment (AC1) · 7 Sep 2016

We thank the referee for the comments and suggestions. Our response and the corresponding changes are listed below (in blue wording).

Tham et al. present a comprehensive set of measurements and analysis focusing on ClNO2 formation at a ground site in Northern China. I thought this was a well written manuscript which should be published after my suggestions below have been considered.

Minor issues.

**1.** pg 2 line 12. (R5) please use the proper chemical symbol for a reversible reaction (the symbol used denotes resonance)
**Response**: The symbol "↔" in R5 has been changed to "⇌" to represent reversible reaction.

**2.** pg 5 - section 2.2. What were the response factors for N2O5 and ClNO2 at m/z 235 and 208 when the Corona discharge and the 210Po were used?
It may be worthwhile to add more detail about the calibration here, and add a figure of an example calibration sequence to the supplemental. The Wang et al. (2016) describes results from a different study, where there was a CRDS N2O5 instrument.

**Response**: The average response factor of 1.11±0.23 pptv/Hz for 235 $m/z$ ($N_2O_5$) and 1.10±0.11 ppt/Hz for 208 $m/z$ ($ClNO_2$) when the corona discharge was used, while the average response factor was 1.32±0.35 pptv/Hz for 235 $m/z$ and 1.40±0.28 pptv/Hz for 208 $m/z$ were determined when the radioactive source was used. This information has been added into the main text. An additional figure has been added into the supplemental to show the example of calibration for $N_2O_5$ and $ClNO_2$ when using the corona discharge and the $^{210}$Po (Figure S3).

[Figure]

Figure S3. An example of sensitivity for $N_2O_5$ and $ClNO_2$ against the relative humidity when (a) corona discharge and (b) $^{210}Po$ were used. The solid line represents the curve fits of the data.

Was the humidity in the CIMS inlet controlled?

The humidity in the CIMS inlet was not controlled. As mentioned in the text, we did standard addition of $N_2O_5$ through the inlet every 3 hours to monitor the sensitivity change of $N_2O_5$ due to ambient changes (i.e. RH and aerosol loading).

I am concerned about the measurement of N2O5 using m/z 235. Can you comment on potential interferences arising from clustering of iodide with organic acids?

There is little information in the literature on the potential interference of 235 $m/z$ regarding the clustering of iodide with organic acids/organics. According to a personal discussion with Y. Chao (from University of Helsinki), in their laboratory measurement with iodide time of flight (ToF)-CIMS, one of the organo-nitrate peaks ($C_8H_{13}O_7N$) indeed located within 235 $m/z$. The ambient concentration of this species should be at sub-ppt level, so we believe that our 235

*m/z* signal should be due to $N_2O_5$ and not largely affected by the organic molecules. In addition, the organo-nitrates is expected to peak in the daytime due to photochemistry, but the 235 *m/z* did not show significant signals in the midday or late afternoon (as shown by the diurnal pattern in Figure 4 in the text).

Is m/z 210 consistent with the relative isotopic abundance of 37Cl?

The 208 *m/z* and 210 *m/z* are consistent with the relative isotopic abundance of chlorine. Plot of 208 *m/z* and 210 *m/z* yields a slope of 0.31 which is near the theoretical value of isotopic chlorine of 0.32. This additional information has been added into the main text and supplement.

[Figure]

Figure S4. Scatter plot of 210 *m/z* against the 208 *m/z*.

**3.** pg 7, line 30 - the $ClNO_2$ cross-sections were remeasured in 2008 by Ghosh et al. (JPC A 116, 6003 (2012)). Please indicate which cross-sections were used in this work.
**Response**: The cross-sections used in the photolysis frequency of $ClNO_2$ for the current analysis was based on the recommendation of Jet Propulsion Laboratory (JPL; Sander et al., 2006). This information has been added into the main text.

Reference:
Sander, S. P., Friedl, R., Golden, D., Kurylo, M., Moortgat, G., Wine, P., Ravishankara, A., Kolb, C., Molina, M., and Finlayson-Pitts, B.: Chemical kinetics and photochemical data for use in atmospheric studies evaluation number 15, JPL Publication 06-2, 2006.

**4.** pg 9 line 29. The homogeneous hydrolysis rate by Wahner et al. is likely incorrect (see, e.g., Brown et al., Science, 2006). Consider omitting it.
**Response**: Thanks for the suggestion. We have omitted the homogeneous hydrolysis rate in the analysis.

**5.** pg 11 line 15. His last name is spelled Riedel.
**Response**: We have corrected the typo.

**6.** pg 12, equation (6). There may also be "loss" of ClNO2 due to entrainment upwards from the residual layer (not just downward mixing). Hence, the levels in the residual layer could be higher than calculated here.

**Response**:

Good point. We agree with the reviewer that there could be upward diffusion of $ClNO_2$ from the residual layer (RL). But the possibility of the upward diffusion is much less than that of the downward diffusion, considering that the mixing between PBL and free troposphere (i.e. the upward diffusion from RL to free troposphere) is much less efficient than the mixing within the PBL (i.e. the downward diffusion from RL to surface). Therefore, in our study, we only considered the downward diffusion of $ClNO_2$ from the RL to the surface to estimate the $ClNO_2$ concentration in RL, and the estimated value is subject to slight underestimation.

We have added a sentence into the main text:

'*The estimated $ClNO_2$ concentration in RL may subject to underestimation due to the omission of the upward diffusion of $ClNO_2$ in RL to the free troposphere.*'

**7.** pg 12 line 19 "This result suggest that elevated ClNO2 may always present in the residual layer of this region." One cannot logically conclude from some observations to "always" as there may be the odd exception. Suggest rephrasing to "frequently" or similar.

**Response**: The word 'always' was rephrased to 'frequently'.

**8.** pg 15 line 27- many references are incomplete (missing doi, volumes, page numbers, etc.).

**Response**: All of the references have been revised.

**9.** pg 22 (Table 1). Please state the uncertainties for each of the measurements.

**Response**: The uncertainty of each measurement have been added to the Table 1.

**10.** General. There are a few minor grammatical errors scattered throughout the document. I would suggest asking a native English speaker to read through the manuscript a couple of times and make corrections where warranted.

**Response**: Thanks for the suggestion. The grammatical errors in the manuscript have been corrected.

---

## Author Comment (AC2) · 7 Sep 2016

We thank the reviewers for their attention to this manuscript. We have made nearly all of the suggested changed and/or clarifications. Our response is in blue wording.

Summary: Tham et al. present a novel set of measurements of ClNO2, N2O5, alongside supporting observations of select trace gases and aerosol. The observations provide new insight on under sampled regions of the atmosphere, particularly with respect to molecules that are recently emerging as being important for atmospheric oxidation.

The analysis follows prior work in this area conducted in the US and Europe and is well founded in the observations. I have a few select comments that should be addressed prior to publication. There are also a series of grammatical errors that should be addressed before the paper makes its way to publication in ACP.

Specific Comments:

**1.** Page 3 lines 1-5: Perhaps discuss in terms of the ClNO2 photolysis lifetime instead of concentrations following sunrise.
**Response**:
Thanks for pointing out. The sentence was rephrased to show the lifetime of ClNO$_2$ due to photolysis after sunrise.

**2.** Page 5 line 10: The use of I- ion chemistry in extremely polluted regions such as this is limited and it is perhaps likely that other atmospheric compounds contribute to the observed signals at 235 and 208 m/z. A few quality control questions: 1) What is the 208/210 ratio for ClNO2 detection, is this consistent with the natural abundance of Cl isotopes? and 2) is there any signal intensity during the daytime (midday / late afternoon) at 235 and 208/210 m/z that would indicate a contribution from other molecules at these masses?
**Response**: The ratio of 208 *m/z* and 210 *m/z* (from the plot below) yields a slope of 0.31 which is consistent with the theoretical value of isotopic chlorine of 0.32. This additional information has been added into the main text and supplement (Figure S4).

[Figure]

Figure S4.  Scatter plot of 210 *m/z* against the 208 *m/z*.

No significant signal intensity was observed in the midday or late afternoon as shown in the diurnal pattern of $N_2O_5$ and $ClNO_2$ in Figure 4 of the main text.

**3.** Page 7 line 31: What is meant by the "physical loss rate of the unmeasured species was set as the 6 h lifetime for the mixing height of 1000 m." Is deposition included in these models?
**Response**: Yes, deposition was included in the MCM model. This is an item for the non-chemical loss of species either through deposition or mixing in the model. The phrase means a lifetime with respect to a physical first order loss of 6 h which equals to a deposition velocity of $V_d = 4.63$ cm $s^{-1}$ in a 1000 m deep boundary layer.

In order to be more specific, we have rephrased the sentence in the text to:
*The lifetime of the unmeasured species with respect to physical first order loss rate was set as 6 h which equals to a deposition velocity of 4.63 cm $s^{-1}$ in a 1000 m deep boundary layer.*

**4.** Page 10 line 6: Converting the computed lifetimes to reactive uptake coefficients based on measured Sa would be a helpful addition as the community is well calibrated to that language. It would also be helpful to include specific values for the ClNO2 yields that best fit the observations.
**Response**: We agree with the reviewer. Specific values for the reactive uptake coefficients and $ClNO_2$ yields have been added into the main text as following:

*... This gives larger $N_2O_5$ uptake coefficient ($\gamma$) of 0.030 in the megacity case compared to 0.014 in the campaign average (estimated from equation 3, where $c_{N2O5}$ is the mean molecular speed of $N_2O_5$).*

$$k(N_2O_5)_{het} = \frac{1}{4} c_{N2O5} S_a \gamma \qquad\qquad (Eq\ 3)$$

*... The $ClNO_2$ yield that best fit the observations can be estimated by dividing the $ClNO_2$ concentration over the integrated amount of $N_2O_5$ uptake loss, as shown in equation (4).*

$$\phi = \frac{[ClNO_2]}{\int k(N_2O_5)_{het}[N_2O_5]\ dt} \qquad\qquad (Eq\ 4)$$

*Comparable average $ClNO_2$ yield of 0.30 and 0.35 are found in the campaign average and megacity case, respectively.*

**5.** Page 11 line 10: What is the accuracy in the measured surface area? Is the surface area reported here dry or wet? If you need a factor of three change in gamma(N2O5) to match the data, is that within the uncertainty in Sa? Especially given that a growth factor may be needed to convert the measured dry to the relevant ambient Sa.
**Response**: We need to clarify that the $S_a$ reported in this manuscript is for the ambient condition, not in the dry state. The wet diameter of particles was calculated with kappa-Köhler function based on the measured size-resolved kappa (refer to the main text). We have made slight modification to the wording of the sentence in the main text to make it clearer.

The uncertainty derived from the estimation of size-resolved kappa was estimated at 16% which is consistent with the uncertainties reported for growth factor and kappa (<20%) (e.g. Yeung et al., 2014; Liu et al., 2014; Hennig et al., 2005). So we do not think that the uncertainty of the calculated $S_a$ may contribute to a factor of 2.4 change in the $N_2O_5$ uptake.

*Reference*:

Yeung, M. C., Lee, B. P., Li, Y. J., and Chan, C. K.: Simultaneous HTDMA and HR-ToF-AMS measurements at the HKUST Supersite in Hong Kong in 2011, J. Geophys. Res.-Atmos., 119, 9864-9883, 10.1002/2013JD021146, 2014.

Hennig, T., Massling, A., Brechtel, F. J., and Wiedensohler, A.: A tandem DMA for highly temperature-stabilized hygroscopic particle growth measurements between 90% and 98% relative humidity, J. Aerosol Sci., 36, 1210-1223, 10.1016/j.jaerosci.2005.01.005, 2005.

Liu, H. J., Zhao, C. S., Nekat, B., Ma, N., Wiedensohler, A., van Pinxteren, D., Spindler, G., Muller, K., and Herrmann, H.: Aerosol hygroscopicity derived from size-segregated chemical composition and its parameterization in the North China Plain, Atmos. Chem. Phys., 14, 2525-2539, 10.5194/acp-14-2525-2014, 2014.

**6.** Page 12 line 15: The calculation of RL ClNO2 is very sensitive to the boundary layer height at 5 and 8AM. Are there measurements of this height? Also, what is the accuracy in the WRF calculated nocturnal boundary layer height? It is hard to imagine this is accurate to the values quoted here (50 and 72m).

**Response**: We agree that the calculated $ClNO_2$ concentration in the residual layer depends on the boundary layer (BL) heights used in the study. We used very high spatial resolution (1 km), high temporal resolution (1h) and observational-nudging techniques in the WRF simulation, which shall give more reliable information than the common global operational analysis data which typically has a spatial resolution of 0.5-1 degree (~50 to 100 km) and temporal resolution of 3-6 h. Previous studies have shown that the parameterization option used in our study (the Yonsei University scheme) appeared to generally reproduce the PBL features in various regions. For instance, Hu et al. (2010) compared observed PBL heights at 8 stations in US and simulated ones from WRF using YSU scheme, and found that the relative simulation bias to be about -13% during the early morning (calculated based on the Fig. 7 in Hu et al., 2010). The information on the bias from the previous study has been added into the text.

We want to clarify that the goal here is not simulate the $ClNO_2$ in the residual layer but rather to estimate its rough concentration levels. We recalculated the $ClNO_2$ concentration with different height of boundary layer. Increasing the simulated nocturnal boundary layer height (5AM) by a factor of 2 while remaining the same boundary layer height at 8AM causes a difference of less than 10% in the $ClNO_2$ concentration. A similar result is obtained while doubling the boundary layer height at 8AM and keeping the simulated nocturnal boundary layer height at 5AM. We have added this result in the revision.

*Reference*:

Hu, X. M., Nielsen-Gammon, J. W., and Zhang, F. Q.: Evaluation of three planetary boundary layer schemes in the WRF Model, J. Appl. Meteorol. Clim., 49, 1831-1844, 10.1175/2010JAMC2432.1, 2010.

---

## Author Comment (AC3) · 7 Sep 2016

We thank the referee for the comments and suggestions which help us improve the quality of the paper. We have made all of the suggested changed and/or clarifications.

This paper describes measurements of nitryl chloride and associated species at a site in the North China Plain (NCP), and presents model estimates of the impact of this active chlorine compound on ozone formation in that environment. The measurements are very interesting and the associated analysis makes some important points about ClNO2 in the residual boundary layer of the polluted NCP. The presentation of the work is quite well done, it is concise and well organized and the important aspects are well explained. There are only a few issues for the authors to address to make this paper acceptable for publication.

General Comments

In general the English in the paper is quite good, however there are a number of instances disagreements between the noun and the verb (e.g. singular when it should be plural, etc.). The authors briefly mention measurements of gas-phase HCl, but since this is an important fraction of the chloride available for activation, it deserves more details. Also, the morning time source of Cl atoms will have a corresponding source of HCl, as most Cl + VOC reactions produce HCl.
**Response**: Thanks for the comments and suggestions. The English have been edited.

We agree that the gas-phase HCl can be an important fraction of the chloride aerosol available for activation. This information has been added into the text (section 3.5) to support the chloride availability.

*The presence of gas-phase HCl (mean of 0.78 ppbv) during the night also can continuously replenish the Cl$^-$ aerosol.*

As for the Cl source, our model analysis had been constrained by the HCl measurement and the mean concentration of HCl was shown in the supplement information (Table S2) to indicate the level of HCl at Wangdu. The contribution of HCl to the daily Cl radical production is much smaller than the photolysis of ClNO$_2$, especially in the morning time (Figure 10a). The related sentence has been revised to include this information.

*It shows that photolysis of ClNO$_2$ was the predominant source of Cl in Wangdu compared to the reaction of HCl and OH and photolysis of Cl$_2$.*

Specific Comments:

**1.** Page 2, Line 31: While ClNO2 is not as well studied as N2O5, there are loss mechanisms for
ClNO2 at night. Kim et al., [2014] show that ClNO2 can be deposited on water surfaces. Roberts et al., [2008] showed that ClNO2 can be taken up on low pH surfaces (and will make Cl2). It is fair to say that because of its low aqueous solubility [Sander, 2015], ClNO2 losses are likely much slower than N2O5, and to a first approximation can probably be neglected.
**Response**: Yes, it is possible for the ClNO$_2$ to undergo loss mechanism under certain conditions. Therefore, we have rephrased the sentence into:

*ClNO₂ may subject to some loss processes on water and other surfaces (e.g. Roberts et al., 2008; Kim et al., 2014), but the night-time losses of ClNO₂ are expected to be negligible due to its low solubility (Sander, 2015).*

**2.** Page 4., Lines 8&9. When you say "tropospheric ozone" that implies a broad scale, really you are talking about ozone in the Beijing urban area.
**Response**: Thanks for the suggestion. The word tropospheric has been omitted.

**3.** Page 5, Line 12. Did you see any evidence of Cl2, at the mass of the cluster ion I(Cl2)- ?
**Response**: We did not measure the cluster ion of $I(Cl_2)^-$ in our CIMS setup.

**4.** Page 7, Line 28, "constrained into" is the wrong expression, a model can be 'constrained by' observations.
**Response**: We thank the reviewer for identifying this error. The phrase has been revised in the text.

**5.** Page 14, Line 29, "less' should be 'lesser'.
**Response**: Corrected.

**6.** Figure 2. The yellow color is hard to see.
**Response**: The color in Figure 2 has been changed.

[Figure]

Figure 2

References
Kim, M. J., Farmer, D. K., and Bertram, T. H.: A controlling role for the air−sea interface in the chemical processing of reactive nitrogen in the coastal marine boundary layer, Proc. Natl. Acad. Sci., 10.1073/pnas.1318694111, 2014.
Roberts, J. M., Osthoff, H. D., Brown, S. S., and Ravishankara, A. R.: N2O5 oxidizes chloride to Cl2 in acidic atmospheric aerosol, Science, 321, 1059., 2008.

Sander, R.: Compilation of Henry's law constants (version 4.0) for water as solvent, Atmos. Chem. Phys., 15, 4399-4981, 10.5194/acp-15-4399-2015, 2015.